# The Preparation and Wear Behaviors of Phenol–Formaldehyde Resin/BN Composite Coatings

**DOI:** 10.3390/polym14194230

**Published:** 2022-10-09

**Authors:** Chao Zang, Yaping Xing, Tingting Yang, Qi Teng, Jinming Zhen, Ran Zhang, Zhengfeng Jia, Weifang Han

**Affiliations:** School of Materials Science and Engineering, Liaocheng University, Liaocheng 252059, China

**Keywords:** phenol–formaldehyde resin/BN composites, wear, coating, heat treatment

## Abstract

Phenolic-matrix composites possess excellent synergistic effects on mechanical and tribological properties and can be used in the aerospace, medical, and automobile industries. In this work, a series of phenol–formaldehyde resin/hexagonal boron nitride nanocomposites (PF/BNs) were in situ synthesized using an easy method. PF/BN coatings (PF/BNCs) on 316L steels were prepared through a spin-casting method. The wear behaviors of these PF/BNCs were investigated by dry sliding with steel balls. The percentage of BN, the thickness of the coating, and the heat treatment temperature affected the coefficients of friction (COFs) and wear rates of these coatings. After heat treatment at 100 °C, the tribological properties of the PF/BNCs were remarkably improved, which might be attributed to both the transformation of carbon on the worn surfaces from C-O/C=O into C=N, carbide, and other chemical bonds and the cross-linking of the prepolymers.

## 1. Introduction

Polymers possess many excellent advantages, such as self-lubricity, high resistance to wear and corrosion, lightweight, rheological properties, and so on, which can be used in many applications, including the aerospace, medical, health, and automobile industries [1,2]. As tribo-materials, polymers also have some shortcomings, such as low strength and poor thermal stability. Polymer-based composites are usually designed to improve tribological properties. In particular, phenol–formaldehyde resin possesses many excellent properties, such as good thermal stability, good resistance to wear, high char yield, and structural integrity, with wide applications in coatings, anti-wear materials, thermal insulation materials, and aeronautic materials [3,4,5]. Many papers have reported that phenolic-matrix composites exhibit excellent synergistic effects on mechanical and tribological properties. Different reinforcements, such as glass fibers/fabric, carbon fibers [6], ceramic nanoparticles, graphene, and soft metals, have been used to promote tribological properties [7,8]. Suitable interfacial bonding, such as surface modification, including acid etching, plasma bombardment, and chemical grafting, is needed to improve the tribological properties of phenolic composites [6,7,8].

Two-dimensional materials are commonly used as reinforcements because of their weak interlayer structures [1,9,10]. Xiao et al. found that the excellent anti-wear capabilities of polyimides with BN were attributed to their good hardness and self-lubricating ability. However, the aggregation of BN increases the roughness, and the particles can be easily pulled out [9]. Combining physical interactions with chemical interactions between reinforcements and resin can improve the tribological behavior of polymer composites [8]. Many methods, such as the introduction of an interfacial linker and plasma treatment, provide strong interlock forces and chemical interactions in the interphase [6,11].

In this work, BN nanosheets were exfoliated by an ultrasonic method in a NaOH solution (named BN-OH), which was transferred to a formaldehyde solution for the fabrication of PF/BN composites through an in situ synthesis method. The films were prepared by the spin-coating technique on a sheet of 316L steel and heat-treated in an argon environment. The results show that coatings heat-treated at 100 °C possess better tribological behavior than those treated at other temperatures. In addition, the friction mechanism of the composite coatings was also investigated.

## 2. Experimental Section

### 2.1. Materials and Preparation

Formaldehyde, h-BN, phenol, NaOH, and ammonium hydroxide were purchased from Yantai Yuandong fine chemical Co., Ltd., China (see Appendix A). All reagents were analytically pure and used without further treatment.

Firstly, h-BN (5.0 g) was transferred to a NaOH solution (concentration: 5 mol/L; volume: 100 mL) with agitation for 30 min and then ultrasonicated to obtain the exfoliated hydroxyl BN powders (BN-OH). Different contents of BN-OH (0.025, 0.05, and 0.1 g) were added to the mixture of formaldehyde (3 mL) and phenol (2 mL) with vigorous agitation at 80 °C. Furthermore, 0.5 g of NH_3_·H_2_O was dropped into the solution to provide an alkaline environment while vigorously stirring for 40 min to obtain the target products (named PF/BNs_0.5%_, PF/BNs_1.0%_, and PF/BNs_2.0%_, respectively). Next, the reactant was spun onto polished 316L steel surfaces at a rotation speed of 1200 rpm for 12 s. The coatings were obtained after being spun three times. The final product was treated in an incubator at 80 °C for 30 min. Finally, steel blocks coated with PF/BNs were heat-treated in a tubular furnace at 100 or 300 °C.

### 2.2. Characterization

HR-TEM (JEM-2100, Japan), AFM (SPA-300HV, Japan), and FE-SEM (SIGMA 500/VP, ZEISS, Oberkochen, Germany) combined with EDXA (Kevex Sigma, Goleta, CA, USA) were used to investigate the microstructures and morphologies of the composite and coatings. XRD (Bruker D8, Billerica, MA, USA), FT-IR (Bruck IFs66v, Billerica, MA, USA), and XPS (ESCALAB Xi+, England) were used to investigate the microstructures of the PF/BNs and worn surfaces. A simultaneous thermal analyzer (NETZSCH STA499, Selb, Germany) was used to investigate the thermal properties of PF/BNs in a N_2_ atmosphere.

### 2.3. Tribological Experiments

A CFT-I tribometer was used to investigate the tribological properties of the samples with a load of 3 N at frequencies of 30, 60, 90, 120, and 150 Hz with a sliding distance of 5 mm for 15 min. AISI 52100 steel balls (Ø: 3 mm) were cleaned using ultrasound before being used as the stationary upper counterparts.

A D-100 profiler (KLA Tencor, USA) was used to measure the area of the profile of scratches on the steel block, and the wear track volumes were counted by multiplying the area of the profile of the scratches by the sliding distance. The wear rates were calculated according to the following equation:W = V/(F × L)(1)

The wear rate (mm^3^ N^−1^ m^−1^), wear track volume (mm^3^), applied load (N), and sliding distance (m) are expressed as W, V, F, and L, respectively [12].

## 3. Results and Discussion

### 3.1. Characterization

Figure 1 exhibits the FE-SEM images of PF, BN-OH, and PF/BNs. The surface of the phenolic is relatively smooth (see Figure 1a) [13]. Gel permeation chromatography (Agilent1260HTGPC) and nuclear magnetic resonance spectrometry (JNM-ECZ400S) revealed that the average molecular weight (Mn) is about 514, with the functional groups Ph, -CH_2_-O-CH_2_-, and Ph-CH_2_-Ph, respectively (see Appendix A). Figure 1b shows the BN sheets after ultrasonic treatment; the thickness of the BN sheets is less than 100 nm. It is obvious that the BN-OH edges are revealed and folded, and a lamellar structure is observed. Figure 1c,d show that BN-OH nanosheets are inlaid into the phenolic matrix. The excellent compatibility between PF and BN is possibly attributable to the exfoliation and hydroxylation of BN sheets.

Figure 2 shows the EDS analyses of the PF/BNs, which shows the elements B, N, C, and O derived from BN and the phenolic in the PF/BNs.

The TEM images of PF/BNs show that PF and multi-layer BN sheets are cohesively composited (see Figure 3), and the thickness of the BN sheets is less than 100 nm. The floccules in PF might be the grains in PF.

Figure 4a shows the XRD curves of PF, BN-OH, and PF/BNs. The diffraction peaks at about 26.7, 41.5, 50.1, and 55.1° could be attributed to the (002), (100), (102), and (004) planes of BN (JCPDS Card # 00034-0421), respectively [14]. The intensity of the (002) peak decreased after being composited with PF, which could be attributed to the decreased degree of crystallinity [15]. The broad peaks at about 19.1° could be related to the crystallized carbon chain in the PF polymer [16], which is consistent with the TEM analysis.

The FT-IR curves of PF and PF/BNs are shown in Figure 4b. The peaks at about 1600 and 1450 cm^−1^ are assigned to the C=C vibration [17]. The absorption peaks at about 1015 and 1250 cm^−1^ are assigned to the C-O vibration of aliphatic groups and methylol aromatic groups, respectively [16]. The peaks at about 3400 cm^−1^ might be assigned to the -OH groups of PF and BN-OH [13]. The peaks at about 820 and 1383 cm^−1^ might be attributed to the stretching and bending vibrations of B-N [17]. A series of peaks from 2300 to 2990 cm^−1^ could be assigned to the stretching vibration of C-H in the polymer structure [16,18].

The TGA curves of BN-OH, PF, and PF/BNs are shown in Figure 4c. Both PF and PF/BNs exhibit substantial weight loss from approximately 135 to 200 °C, possibly due to the cross-linking polymerization of the PF prepolymer and dehydration [19,20]. The second inflection points of TG curves are at about 380 °C, possibly attributed to the carbonization of phenol–formaldehyde resin [21]. The weight loss of PF/BNs is less than that of pure PF, proving the existence of BN in composites.

The XPS spectra of the PF/BN samples are shown in Figure 5. The spectra of B1s and N1s reveal that the peaks at 190.0 and 398.8 eV can be assigned to B-N bonds (Figure 5b,c) [22]. The binding energy at about 192.5 eV might be attributed to B-O bonds. For the N 1s spectrum, the peak at 401.3 eV can be attributed to C-N bonds [23]. The high-resolution C1s curve of PF/BNs in Figure 5d can be deconvoluted into four peaks at 284.2, 285.7, 286.2, and 291.0 eV, corresponding to C-C, C=C, C-O, and π-π chemical bonds, respectively [24,25]. The O 1s curve of PF/BNs in Figure 5e can be deconvoluted into four peaks at 530.6, 532.0, 532.7, and 533.1 eV, corresponding to -OH, C=O, C-O, and N-O bonds, respectively [15]. The above information proves that PF/BNs were successfully prepared, and BN was hydroxylated.

Figure 6 shows the SEM and AFM analyses of pure PF and PF/BNC coatings. Comparing the images of the PF coating, there are some bright features on the surfaces of PF/BNCs, which are attributable to BN particles (Figure 6a,b). The thickness of the PF/BNCs is about 30 μm (Figure 6c), and the roughness of the PF/BNCs is about 1.597 nm (see Figure 6d).

### 3.2. Tribological Behavior

Figure 7 shows the wear-rate–frequency curves and COF–frequency curves of PF coatings and PF/BNCs on steel discs sliding against AISI steel balls. The wear rates of PF coatings are lower than those of the steel discs at frequencies of no more than 120 Hz, as shown in Figure 7a. The friction coefficient of steel discs decreases after being coated with PF at all frequencies. At a frequency of 90 Hz, the average COFs of steel discs drop from 0.52 to 0.45 after PF coating (Figure 7b). The wear rates and COFs of PF/BNCs with different percentages of BN were also investigated. BN (1.0 wt%) can decrease the wear rates and COFs at all frequencies. At a frequency of 90 Hz, the wear rates and COFs of the PF/BN_1.0%_-coated sample are 0.26 × 10^−4^ (mm^3^/N·m) and 0.357, which are much lower than those of 0.30 × 10^−4^ (mm^3^/N·m) and 0.45 for the pure-PF-coated sample, respectively.

Figure 8 shows the SEM images and mapping images of the worn surfaces of steel, PF coatings, and PF/BN_1.0%_ coatings sliding against steel balls at 90 Hz. The steel discs are badly worn with deep furrows and many abrasions, which indicate that serious abrasive wear occurred during sliding. The worn surface of the PF-coated sample displays much smoother wear than that of pure steel blocks [9]. Furthermore, the PF/BNCs show even less wear than the PF coatings, which is consistent with the wear rate and COF values (see Figure 7) [26]. The EDS mappings of the worn surfaces of PF/BNCs (Figure 8f) show that B, N, O, C, and Fe elements exist on the worn surfaces (Figure 8g–k), from which it can be deduced that the coating of the tribo-films containing the above elements improves the tribological properties [15].

To investigate the tribological properties of PF/BNCs with different thicknesses, the wear rates and COFs of PF/BNs_1.0%_ coated 1 time, 3 times, and 5 times were investigated. The PF/BNCs coated 3 times possess a lower wear rate and COF than those coated 1 and 5 times. The optical images of the worn surfaces show that the worn surfaces of the films coated 3 times are much smoother than those of the other samples, which is inconsistent with the wear rates and COFs (see Figure 9a). The COF changes during sliding for composite films with different thicknesses are shown in Figure 9b. The curves violently fluctuate after the process is stable for a few minutes. The optical images of the worn surfaces show that the direct increase in COF indicates the destruction of the coatings. The films coated 3 times have a longer period of stable COFs.

On the other hand, the PF/BNs_1.0%_ composite coatings were heat-treated in a tubular furnace at temperatures of 100 °C and 300 °C. The tribological properties of the PF/BN composite films after heat treatment were also investigated at a frequency of 90 Hz for 15 min (Figure 10). The wear rate of the PF/BNCs decreased from 2.59 × 10^−4^ to 1.58 × 10^−4^ mm^3^/N·m after heat treatment at 100 °C, which is in line with the cross-sectional profiles and SEM images of the worn surfaces. At the same time, the COF also decreased from 0.38 to 0.36 after heat treatment at 100 °C. However, after heat treatment at 300 °C, the wear rates and COFs of the films were higher than those at 100 °C. So, it can be deduced that the PF/BNCs heat-treated at 100 °C possess better anti-wear and friction-reducing abilities than other samples.

### 3.3. Discussion

Many investigations have been conducted to improve the friction and wear mechanisms of phenolic resin and its composites. Zhang et al. found that a PF–graphene composite coating exhibited enhanced tribological properties under all tested conditions, which could be attributed to the increased interfacial interaction between graphene and phenolic resin [20]. It is interesting to find tribological mechanisms from other possible aspects. To determine the anti-wear and friction-reducing mechanisms of the PF/BNCs, the XPS spectra (elements C, O, B, N, Fe) of the wear scratches of PF/BNCs before and after heat treatment at 100 °C were investigated (see Figure 11). The high-resolution C1s spectrum of the worn surfaces of PF/BNCs is divided into the peaks of C-C (283.7 eV), carbon (284.7 eV), C=C (285.2 eV), C-O (286.4 eV), and C=O (285.7 eV) [27]. On the other hand, the high-resolution C1s spectrum of the worn surfaces of PF/BNCs heat-treated at 100 °C is divided into the peaks of C=N (285.2 eV), carbon (284.8 eV), C-C/C-H (284.1 eV), carbide (282.8 eV), and other groups (283.4 eV). The above analysis potentially proves that the C element on the worn surfaces of the coatings after heat treatment is transferred from C-O/C=O and C-C to C=N, carbide, and other chemical bonds, which is possibly attributed to the cross-linking of the PF prepolymers (see Figure 11a,f) [12]. Comparing the O1s curves before and after heat treatment, it is observed that the curves from the worn surfaces after heat treatment are noisier than before heat treatment, possibly due to the decreased oxygen contents after heat treatment (see Figure 11b,f). The peaks at about 530.3, 531.0, 531.8, 532.3, and 533.0 eV are attributed to the chemical bonds of Fe-O, C=O, -OH, C-O, and N-O, respectively [15]. The decreased peak areas of N-O and C-O after heat treatment might be because of the cross-linking of prepolymers and the loss of H_2_O and other micromolecules at the same time [28,29]. The weak peaks of N1s of the worn surfaces are identified as C-N and B-N at 401.3 and 398.7 eV, respectively (see Figure 11c). After heat treatment, the B-N bonds are divided into nitride and cyanides, respectively [30]. The peak of B1s is too weak to be observed due to its low concentration. The peaks of Fe2p are also very weak, which might be attributed to the existence of PF/BN tribo-films. Furthermore, weak Fe-O bonds are found on the worn surface. It can be concluded that tribo-films containing C, O, N, and Fe are formed on the worn surfaces of the PF/BNCs, which can effectively protect the steel from damage [12]. Comparing the TGA analyses of PF/BN, it can be concluded that the cross-linking of the prepolymers occurs during heat treatment, and carbon and nitrogen are transferred to C=N bonds, carbides, nitride, and cyanides at the same time, which improves the tribological properties of these coatings.

The mechanical properties were tested to further understand the friction and wear mechanisms. The hardness and toughness of materials play an important role in the investigation of the friction and wear mechanisms of coatings [31]. Furthermore, the indentation hardness (HIT), indentation elastic modulus (EIT), and load–depth curves of the PF/BNCs were measured using a Nanoindentation apparatus (TTX-NHT2, Swit) with a diamond indenter (B-U 04, Berkovich) for a maximum load of 0.9 mN and an approach distance of 2000 nm with an approach speed of 2000 nm/min, respectively. Interestingly, the PF/BNCs heat-treated at 100 °C possess a lower HIT and a higher indentation EIT than those without heat treatment and those heat-treated at 300 °C, respectively (see Figure 12a–c). The adhesive forces of the PF/BNCs with and without heat treatment were also investigated using a scratch tester (MFT 4000, China) with a loading force of 30 N, a speed of 30 N/min, and a sliding distance of 5 mm. The adhesive force of PF/BNCs heat-treated at 100 °C is about 7.8 N, much higher than those of coatings without heat treatment and those heat-treated at 300 °C (see Figure 12d). The corresponding optical images of the wear scars of the coatings without heat treatment and those heat-treated at 300 °C show that large debris was peeled off at the first point. The samples are more degraded than those treated at 100 °C. The COF–loading-force curves also show that PF/BNCs heat-treated at 100 °C have more stable COFs than other samples (see Figure 12e). The above analysis proves that PF/BNCs heat-treated at 100 °C possess an improved EIT with a lower COF and a higher adhesive force between the coating and steel substrate than the other samples, which improves the anti-wear and friction-reducing abilities of the PF/BNCs [32].

The XPS analysis of the PF/BN powders heat-treated at 100 or 300 °C reveals that the percentages of C-O and C=O bonds decrease as the heat treatment temperature increases to 300 °C (see Appendix A). Furthermore, the ammonium salt was transferred to nitride when increasing the heat treatment temperature to 300 °C (see Appendix A) [33]. The B1s analysis reveals that the B-O bonds become weakened after heat treatment at 300 °C (see Appendix A). From the TGA analysis and XPS analysis, we can deduce that carbon and nitride were formed during the heat treatment at 300 °C, which was also verified by contact angle measurements (see Appendix A) [6]. The samples with carbon and nitride possess excellent hardness but have a relatively low EIT (see Figure 12), which might cause them to easily peel off during sliding because of their brittle properties [34].

There are two explanations for the mechanism of the PF/BNCs. Tribo-films containing C, O, B, N, and Fe are formed on the worn surfaces of the PF/BNCs. The C element on the worn surfaces of the coatings after heat treatment at 100 °C is transferred to C=N, carbide, and other chemical bonds, which can effectively protect the steel from damage. On the other hand, the cross-linking of the prepolymers improves the elastic modulus and adhesive forces between the film and substrate, improving the tribological properties of PF/BNCs (see Figure 13) [6,35].

## 4. Conclusions

In this work, PF/BNCs were synthesized using an eco-friendly method. The percentage of BN, the thickness of the coating, and the heat treatment temperature affect the COFs and wear rates of these coatings. After heat treatment at 100 °C, the wear rates and COFs of the PF/BNCs were reduced from 2.59 × 10^−4^ mm^3^/N·m and 0.38 to 1.58 × 10^−4^ mm^3^/N·m and 0.36, respectively. However, after heat treatment at 300 °C, the wear rates and COFs of the films were higher than those at 100 °C. The C element on the worn surfaces of the coatings after heat treatment at 100 °C was transferred to C=N, carbide, and other chemical bonds, which can effectively protect the steel from damage. Furthermore, the cross-linking of the prepolymers increased the elastic modulus and adhesive forces between the film and steel, which improved the tribological properties of PF/BNCs.

## Figures and Tables

**Figure 1 polymers-14-04230-f001:**
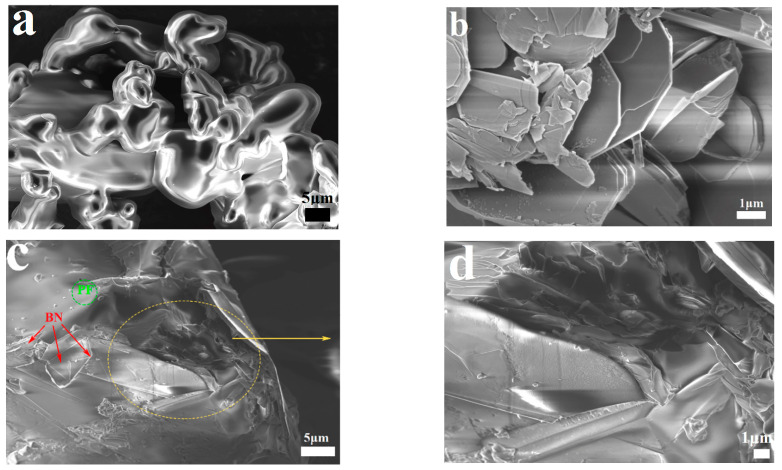
FE-SEM images of (**a**) PF, (**b**) BN-OH, and (**c**,**d**) PF/BNs.

**Figure 2 polymers-14-04230-f002:**

SEM image (**a**), EDS elemental mappings (**b**–**e**), and EDS curve (**f**) of PF/BNs.

**Figure 3 polymers-14-04230-f003:**
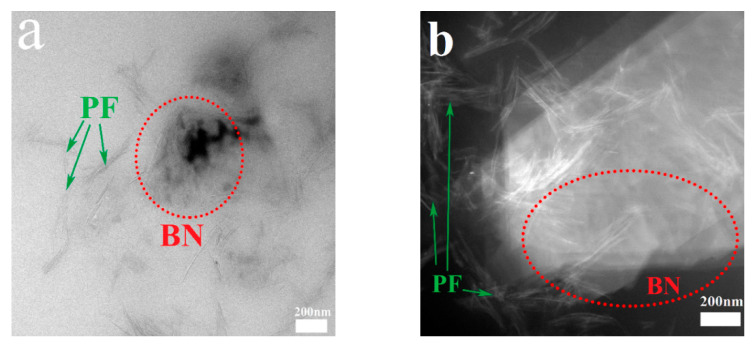
HRTEM images of (**a**,**b**) PF/BNs.

**Figure 4 polymers-14-04230-f004:**
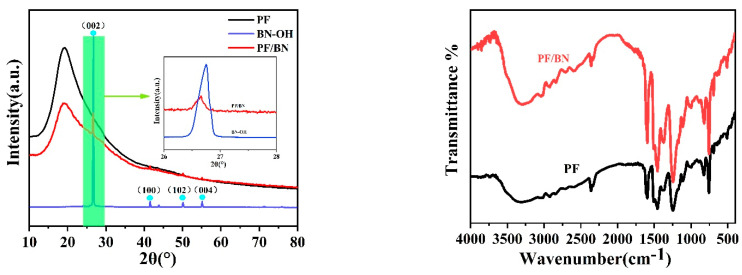
XRD curves (**a**), FT-IR (**b**), and TGA curves (**c**) of PF, BN-OH, and PF/BNs.

**Figure 5 polymers-14-04230-f005:**
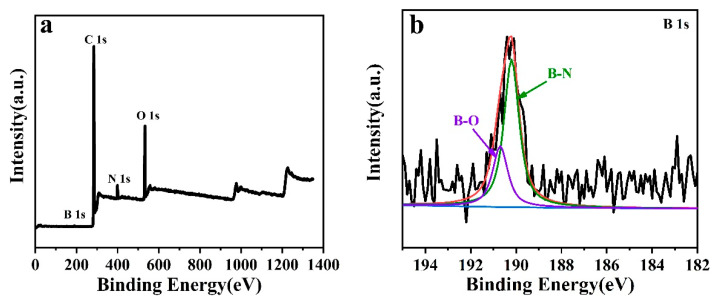
XPS spectrum (**a**) and deconvoluted curves (**b**–**e**) of PF/BNs.

**Figure 6 polymers-14-04230-f006:**
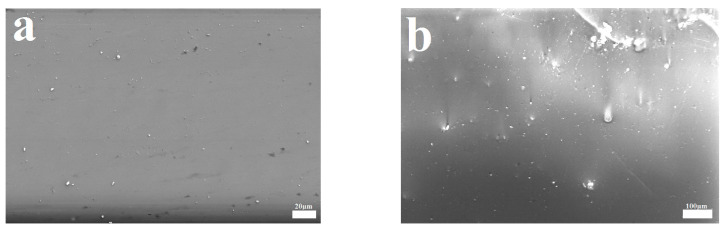
SEM images of PF coating (**a**) and PF/BNCs (**b**), profile map (**c**), and AFM image (**d**) of PF/BNCs.

**Figure 7 polymers-14-04230-f007:**
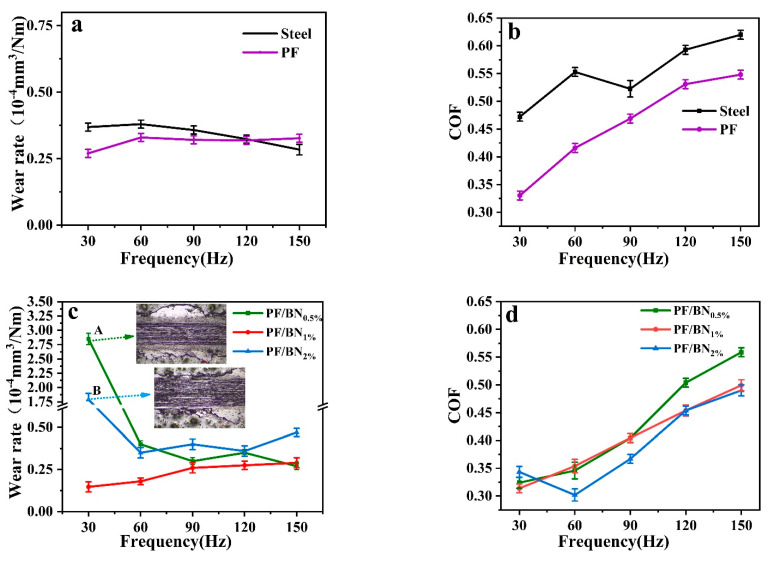
Wear-rate–frequency curves (**a**) and COF–frequency curves (**b**) of steel discs before and after being coated with PF. Wear-rate–frequency curves (**c**) and COF–frequency curves (**d**) of PF/BNCs sliding against steel balls.

**Figure 8 polymers-14-04230-f008:**
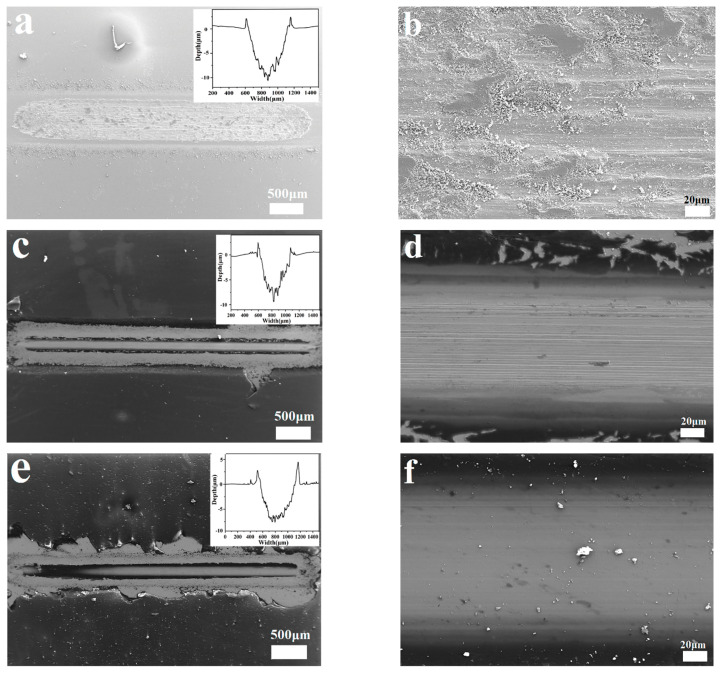
SEM images of the worn surfaces of steel (**a**,**b**), PF coatings (**c**,**d**), and PF/BN_1.0%_ coatings (**e**,**f**) at a frequency of 90 Hz and the mapping images (**g**–**k**) of the image in (**f**).

**Figure 9 polymers-14-04230-f009:**
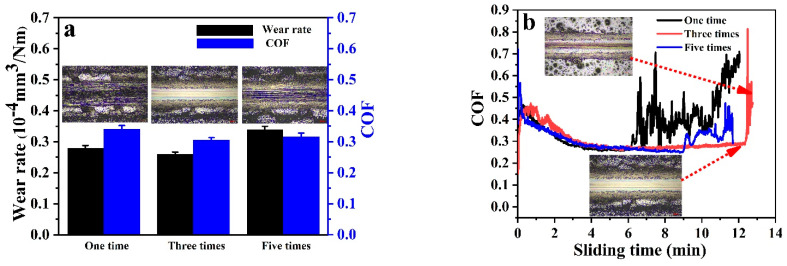
(**a**) Wear rates and COFs of PF/BNs_1.0%_ coatings with different thicknesses at a frequency of 90 Hz for 15 min; (**b**) friction-time curves of PF/BNs_1.0%_ coatings with different thicknesses.

**Figure 10 polymers-14-04230-f010:**
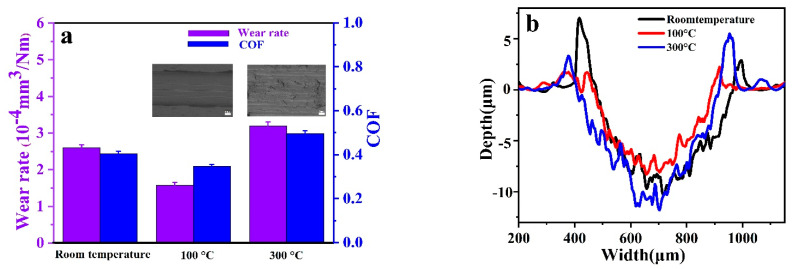
(**a**) Wear rates and COFs of the PF/BNs_1.0%_ coatings with/without heat treatment at a frequency of 90 Hz for 15 min; (**b**) cross-sectional profiles of the worn surfaces corresponding to (**a**).

**Figure 11 polymers-14-04230-f011:**
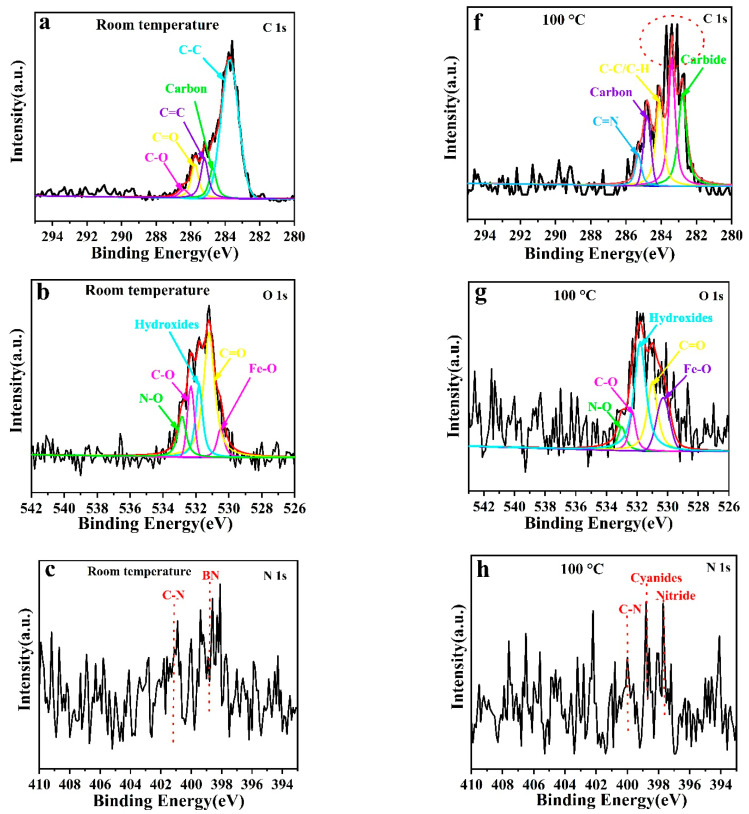
XPS curves for the worn surfaces of the PF/BNCs (**a**–**e**) and those of coatings heat-treated at 100 °C (**f**–**j**).

**Figure 12 polymers-14-04230-f012:**
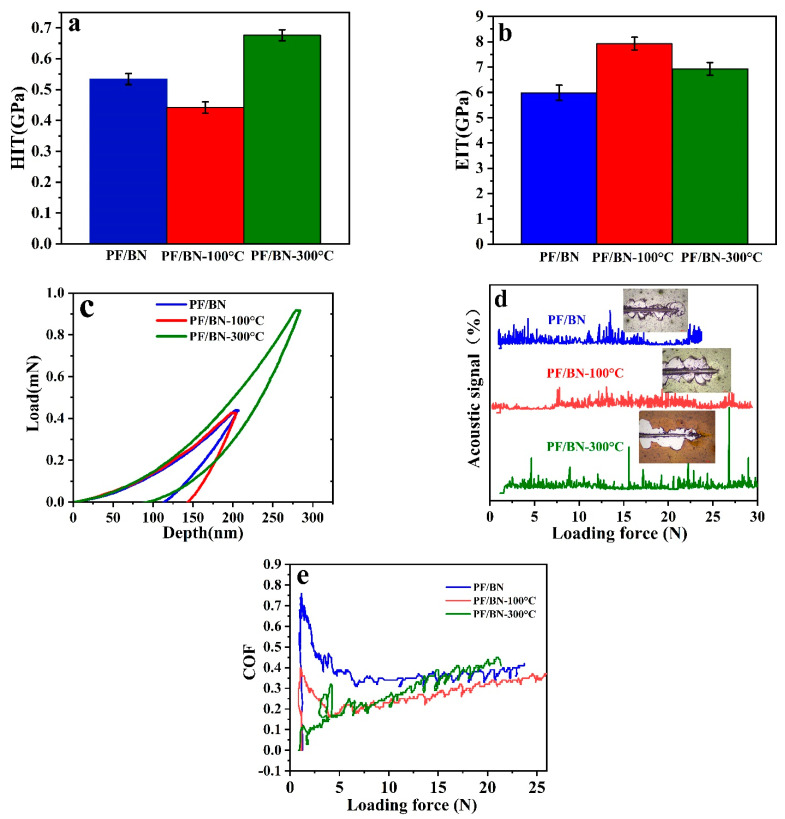
HIT (**a**), load–depth (**b**), ETT (**c**), COF–loading-force (**d**), acoustic signal–loading-force (**e**) curves of the PF/BNCs with and without heat treatment.

**Figure 13 polymers-14-04230-f013:**
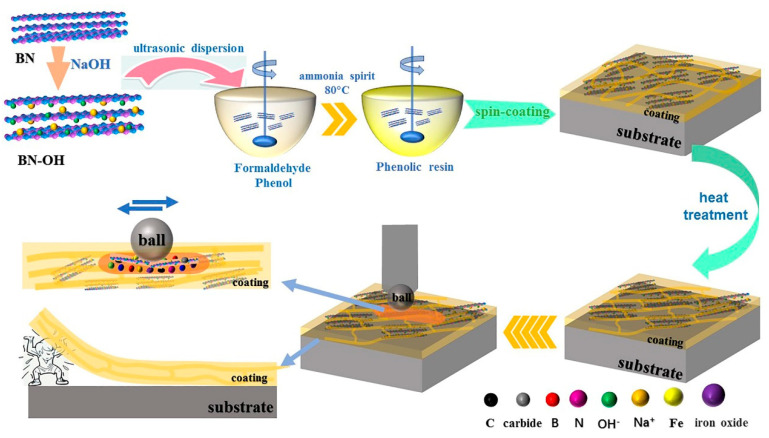
Sketch map of the preparation and tribological mechanism of PF/BNCs.

## Data Availability

Not applicable.

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
