# Peer review of "The Preparation and Wear Behaviors of Phenol–Formaldehyde Resin/BN Composite Coatings"

_polymers, 2022, doi:10.3390/polym14194230_

Round 1

Reviewer 1 Report

Comment 1: Qualitative informations are missing in abstract. Abstract should be concise and the authors need to improve with more specific short results.

Comment 2: Title should be revised and improved.

Comment 3: The novelty of the work should be established.

Comment 4: The purities of the all used chemical products should be mentioned in the Materials and preparation part.

Comment 5: Numbers of equation should be mentioned in the text.

Comment 6: The figures in the manuscript were poor, the author should improve the quality and solution of these figures.

Comment 7: Conclusion is too, conclusion should be revised and improved.

Comment 8: Compare your results with literature ones.

Comment 9: The introduction section should be modified and improved though citing recent references (2021 and 2022) related studies and indicating the novelty of the study compared to the carried works. The following references should added.

ü Polymer Bulletin 76 (9) (2019) 4859-4878 (https://link.springer.com/article/10.1007/s00289-018-2639-9).

ü Composite Structures 262 (2021) 113640 https://doi.org/10.1016/j.compstruct.2021.113640

ü Heliyon 6 (2020) e04187 (https://doi.org/10.1016/j.heliyon.2020.e04187).

ü SN Applied Sciences 1 (2019) 1-9 (https://doi.org/10.1007/s42452-019-0911-8).

ü Journal of King Saud University Science 32 (2020) 235-244 (https://doi.org/10.1016/j.jksus.2018.04.030).

Reviewer 2 Report

The paper presents an interesting approach based on the The preparation, characterization and tribological properties of phenol-formaldehyde resin /BN composite coatings on steel. However, the innovation of the current research work should be further highlighted and emphasized. At the same time, the authors should consider the following comments to greatly improve the quality of the paper.

1. In the abstract, add a final statement that highlights the importance of this research and its possible potentials. Also, introduce the problem in the initial lines of the abstract.

2. The introduction needs to be improved by relating to the mechanics of the studied materials and their mechanical characteristics. The references to be included are: 10.1177/0021998318790093, 10.1016/j.polymertesting.2017.09.009, 10.1016/j.compstruct.2021.114698, 10.1177/0731684417727143, 10.1002/app.46770, 10.1016/j.porgcoat.2022.107015, 10.1177/07316844211051733.

3. Kindly add a table that describes the main physical and chemical properties of the raw materials used in this study.

4. Were the preparation methods described by the authors come in accordance with a certain standard or do they follow previous procedures?

5. For the tribological tests, what was the reason for the specified loading conditions in this research? Do the sliding time, speed and load values relate to a specific application? Why there were five different frequencies imposed in this test matrix?

6. The FE-SEM images in Figure 1 come all in different magnifications for different compositions. How can the comparison be possible? Kindly add the same magnification factor images for all compositions.

7. The elemental analysis in Figure 2.f is not clear, kindly magnify the table or present it in a separate table.

8. For the dot mapping in Figure 2.b and Figure 2.c, what is the difference between the views of B element and N element?

9. The conclusion needs to be modified to summarize the research outcomes in short statements with clear observations.

Reviewer 3 Report

In this work, the author synthesized a series of PF/BNs coatings on steels, and studied the wear behaviors by dry sliding. The coating is characterized by different methods. Then, they systematically investigated the effects of various factors on the friction coefficients and wear rate. These results are important for understanding the tribological properties. Therefore, I suggest a minor revision, with the following questions hopefully helpful for improving the quality.

1. The C element on the worn surfaces of the coatings after heat are transferred into C=N, carbide, and other chemical bonds thus can effectively protect the steels from damage. This is interesting and the authors can further discuss this mechanism.

2. In Page 5, right below Figure 4, there is a Table 4, but I cannot find anything related. Please check that again.

3. Figure 10, why at 300℃ the tribological property becomes worse? Could the authors please provide some speculations or explanations for that?

Round 2

Reviewer 2 Report

The references recommended to be used weren't included in the article. Kindly add them and resubmit.